# **Technical Note: Monitoring streamflow generation processes at Cape Fear**

Flavia Tauro<sup>1</sup>, Andrea Petroselli<sup>2</sup>, Aldo Fiori<sup>3</sup>, Nunzio Romano<sup>4</sup>, Maria Cristina Rulli<sup>5</sup>, Maurizio Porfiri<sup>6</sup>, Mario Palladino<sup>4</sup>, and Salvatore Grimaldi<sup>1,6</sup>

<sup>1</sup>Dipartimento per l'Innovazione nei Sistemi Biologici, Agroalimentari e Forestali, University of Tuscia, Viterbo 01100, Italy. <sup>2</sup>Department of Agriculture and Forestry Sciences (DAFNE), University of Tuscia, Viterbo 01100, Italy.

<sup>3</sup>Dipartimento di Ingegneria, RomaTre University, Rome 00146, Italy

<sup>4</sup>Department of Agricultural Sciences, AFBE Division, University of Napoli Federico II, Napoli 80138, Italy

<sup>5</sup>Dipartimento di Ingegneria Civile e Ambientale, Politecnico di Milano, Milan 20133, Italy

<sup>6</sup>Department of Mechanical and Aerospace Engineering, New York University Tandon School of Engineering, 11201 Brooklyn, NY, USA

Correspondence to: Salvatore Grimaldi (salvatore.grimaldi@unitus.it)

Abstract. Hillslope processes are fundamental for the comprehension of the hydrological response of natural systems. However, their complexity demands real time and continuous observations. In this paper, we assess the feasibility of studying streamflow generation processes at Cape Fear, a "hybrid" hillslope plot at University of Tuscia, Viterbo, Italy. Cape Fear is a  $7 \times 7 \text{ m}^2$  confined soil-filled wood-sided plot, whose water fluxes can be continuously monitored. The plot design is simple, yet

5 versatile to test hypotheses on the hydrological response of hillslope areas. The suitability of the plot for investigating runoff generation and hillslope processes is presented through a demonstrative experiment in the case of a natural rainfall event. A combination of traditional and innovative measurement techniques confirms that runoff onset is due to saturation overland flow. Future studies will address the influence of diverse land covers and spatial pathways evolution on the response at the hillslope scale.

## 10 1 Introduction

Hydrological processes occurring at the hillslope scale highly influence the response of natural catchments (Dunne, 1978). Over the years, considerable efforts have been devoted to the identification and modelling of the mechanisms through which hillslopes store and release water (Janzen and McDonnell, 2015). Among multiple processes occurring at the hillslope scale, filling and spilling of infiltrated rainfall and subsurface stormflow are regarded as the major drivers of hillslope runoff (Spence,

15 2010). However, depending on soil properties, hortonian and saturation overland flow may also be significant contributions to storm hydrographs (McDonnell, 2013).

Hydrological processes at the hillslope scale are complex phenomena that are influenced by diverse environments and can significantly vary in different parts of the same catchment (Beven, 2001). Modelling hillslope dynamics is often extremely challenging, whereby analytical conceptualizations may be inadequate to simulate such complex processes (Janzen and Mc-

Donnell, 2015). Towards this aim, field experiments have been frequently conducted in outdoor plots. Such facilities can be broadly classified in natural instrumented slopes and artificial plots.

Large scale natural slopes are critical toward an improved understanding hydrological processes (Kendall et al., 2001; McGuire et al., 2007). For instance, in (Montgomery et al., 1997), the hydrological response of two densely instrumented zero-order catchments in the Oregon Coast Range is studied. In particular, the response of such unchannelled valleys to natural

- 5 zero-order catchments in the Oregon Coast Range is studied. In particular, the response of such unchannelled valleys to natural and artificial rainfall has been thoroughly investigated with traditional instrumentation and tracer techniques (Anderson et al., 1997a, b). Extensive field experiments have also been conducted in the 41 ha Panola Mountain Research catchment. Field data captured therein have enabled investigations on the role of bedrock topography on subsurface storm flow (Freer et al., 2002), and have led to the formulation of the fill and spill hypothesis (Tromp-van Meerveld and McDonnell, 2006a, b; Klaus
- et al., 2013). Further, the catchment-scale Hydrological Open Air Laboratory (HOAL) in Lower Austria is a 66 ha laboratory (Blöschl et al., 2015), which is an extremely rich testbed for the comprehension of water flow and transport processes.
   While field campaigns at the catchment scale are necessary in hydrology, the complexity of site geometry and uncertainties in boundary conditions and water fluxes mandate long-term and frequent observations by multidisciplinary research teams.
- Alternative to catchment-scale studies, the response of hillslopes has also been investigated by isolating experimental plots 15 in natural watersheds (Bachmair et al., 2012). For instance, in (Brooks et al., 2004), plastic and metal sheeting is utilized to study the response of a  $35 \times 18 \text{ m}^2$  slope in a natural field. Similarly, in (Scherrer et al., 2007), trenches are excavated at numerous  $4 \times 15 \text{ m}^2$  natural hillslope sites to monitor runoff formation during intense artificial precipitation. These studies offer the advantage of experimenting with natural soil structure. However, similar to catchment-scale studies, controlling water fluxes may be challenging.
- Artificial hillslope plots are precious research tools to "bridge the gap between abstract statistical realism and detailed realism" (Janzen and McDonnell, 2015). Most importantly, they offer enhanced controllability of input fluxes, repeatability of investigations, and a priori determined boundary conditions. An outstanding example is the Biosphere 2 Hillslope Experiment, Arizona, where a system of three  $18 \times 33 \text{ m}^2$  hillslopes has been assembled to study the evolution of soil, water, and biota (Hopp et al., 2009; Niu et al., 2014). The design of this unique facility involves several research groups and is expected to open novel perspectives for a unifying hydrological theory at the catchment scale.
  - Most frequently, plot scale studies are conducted in artificially assembled outdoor or indoor facilities of a few square meters. For instance, the effect of vegetation on water fluxes and erosion is investigated in (Wainwright et al., 2000), overland flow times on artificial slopes are studied in (Li and Chibber, 2008), and the effect of vegetation and ground cover on erosion is analyzed through three small scale plots in (Ghahramani et al., 2011). Simple facilities to study overland flow, infiltration, and

soil moisture are also presented in (Esteves et al., 2000; Kampf and Burges, 2007). Generally, such plots present homogeneous

soil and basic monitoring equipment, and they are mostly adopted to test hydrological models.

30

Cape Fear is a "hybrid" outdoor hillslope plot for testing fundamental hypotheses on runoff generation processes. The plot shares similarities with both laboratory facilities and catchment-scale experimental hillslopes. Similar to laboratory boxes, Cape Fear is a confined soil-filled wood-sided plot, whose water fluxes can be controlled and infiltration to deeper soil layers is

5

hampered. On the other hand, similar to natural instrumented slopes, the plot is subject to external agents, such as temperature and rainfall.

The "intermediate" features of Cape Fear offer several advantages for hydrological observations. Firstly, simplified and controllable geometry and morphology allow for easily identifying the relative contribution of hydrological processes taking place in the plot. Further, the facility enables controllability of water fluxes through real time observations with a low cost sensing infrastructure. Sensing equipment can be easily rearranged (for instance, the soil and slope of the plot can be modified, along with the instrument location) according to specific experimental requirements. Finally, the setup is directly exposed to external events, thus allowing for the establishment of hydrological processes found in natural ecosystems.

In this paper, we assess the feasibility of studying streamflow generation processes in the "intermediate" experimental settings of Cape Fear. To this end, we study the response of Cape Fear to external precipitation through a proof-of-concept experiment. In particular, we study the onset of surface runoff in the plot during a natural precipitation event occurred on January 30th, 2014. The response of the plot to a storm is reconstructed based on continuous monitoring of input and output fluxes. Further, the high-visibility surface tracer previously adopted in (Tauro et al., 2010, 2011, 2012c) is utilized to remotely investigate overland flow from captured images. Such image-based monitoring approach has been widely used in preliminary

studies on overland and stream flows (Tauro et al., 2012a, b). Insight from experimental observations is leveraged to identify the physical phenomena governing the response of the hillslope to the precipitation event.

The paper is organized as follows. In Section 2, the natural storm experiment is described as well as the design of the outdoor laboratory and the characterization of its components. In Section 3, experimental observations are reported. In Section 4, the performance of the outdoor plot and possible research directions are discussed. Conclusions are left for future improvements are descent by

20 and remarks.

## 2 Materials and Methods

#### 2.1 Streamflow generation experiment

The feasibility of studying streamflow generation processes at Cape Fear is assessed by conducting an experiment driven by a natural rainfall event. Specifically, we observe and study the plot hydrological regime for a time window of 30 hours, from

- 10:00 on January 29th, 2014 to 16:00 on January 30th, 2014. During such a time, two major precipitation events were forecast. The first event took place in the night between January, 29th and 30th. The second event was anticipated to occur in the afternoon on January, 30th. Prior to the experiment, the hillslope is left uncovered for two weeks to allow stable moisture and infiltration conditions to be established in the plot. During the experiment, input rainfall, output discharge, turbidity, and soil moisture are surveyed and saved every five minutes. Further, we utilize a fluorescent particle tracing system to investigate the
- onset of surface runoff.

## 2.2 Experimental site

Cape Fear is realized in a terrain parcel in the Azienda Agraria at the University of Tuscia, Viterbo, Italy. The plot extends on an area of 7 × 7 m<sup>2</sup>, is constructed out of 40 m<sup>3</sup> of soil, and has an average slope of 17%, see Figure 1(a). The hillslope is assembled by filling a containment structure of wood boards and poles on three sides with soil. The plot is completely
disconnected from the underlying terrain through a waterproof layer. The rather invasive plot assembly allows for a priori controlled geometry and boundary conditions. A system of four rainfall simulators, based on the instrument adopted in (Tauro et al., 2012b) and initially developed in (Riley and Hancock, 1997), is installed in the hillslope to uniformly irrigate the plot. Rainfall is provisioned and regulated through a water feeding system, including a tank, a hydraulic pump, a flow meter, and a buried piping, see Figure 1(b).

The experimental site features a real-time monitoring system, which affords the acquisition of data on input artificial and natural rainfall, output water discharged from the plot, turbidity, soil moisture conditions, and qualitative information on the wetting front propagation. A priori determined plot geometry and boundary conditions allowed for the installation of a relatively meagre number of sensors. Acquired values are recorded and stored by a CR10X Campbell Scientific datalogger. The logger is able to store sets of data recorded at time intervals of five minutes from all the installed instruments for maximum periods of 15 ten days.

As displayed in Figure 1(c), the site includes a working region where undisturbed conditions are maintained. The region is a  $4 \times 4 \text{ m}^2$  square with the rainfall simulators located at its vertices. Around the undisturbed working area, 1.5 m-wide sections are left from the external side of the hillslope to mitigate border effects.

#### 2.2.1 Soil physical and hydrological characterization

- Cores are obtained from the uppermost 20 cm of the plot by driving a steel cylinder vertically into the soil. A hand-operated device is used while excavating by hand the soil around the cylinder to limit disturbances during sampling. In Figure 2, the locations of the soil core samples are indicated in a plane sketch view of the plot. The soil cores are collected using steel cylinders of about 0.07 0.08 m in diameter. In experiments on the suction table (soil cores collected at locations from P02 to P05), steel cylinders are 0.07 m in height. Further, an evaporation experiment is conducted on a 0.15 m core collected at location
- P01. Before performing hydraulic tests, the top of each undisturbed soil core (a thickness of approximately 0.03 0.04 m) is removed and stored for particle-size analyses, organic content measurements, and for measuring soil water retention data in the dry range through pressure plate apparatus.

The particle-size distribution is determined using standard laboratory techniques based on a set of sieves and the soil hydrometer (Gee and Or, 2002). The organic carbon content is determined with the dichromate method (Mebius, 1960). Saturated
water content, θ<sub>sat</sub>, is measured by the gravimetric method (Topp and Ferré, 2002), whereas the saturated hydraulic conductiv-

ity,  $K_{\text{sat}}$ , is measured by the falling-head method (Reynolds et al., 2002).

Table 1 reports the physicochemical and hydraulic properties of the collected soil samples. As often confirmed in the literature, at the small space scale of the plot, the variability of the soil physical properties is relatively small. Specifically, the

15

Table 1. Hydraulic properties of the soil samples.

|                                             | C039 P01 | C376 P02 | C017 P03 | C385 P04 | C320 P05 |
|---------------------------------------------|----------|----------|----------|----------|----------|
| $\rho_{\rm b}({\rm gcm}^{-3})$              | 0.979    | 1.025    | 1.090    | 1.025    | 1.048    |
| Organic C (%)                               | 0.448    | 0.721    | 0.604    | 0.526    | 0.584    |
| Sand content (%)                            | 45.87    | 42.40    | 46.52    | 43.70    | 41.63    |
| Silt content (%)                            | 33.38    | 34.52    | 34.00    | 38.94    | 37.19    |
| Clay content (%)                            | 20.75    | 23.28    | 19.48    | 17.36    | 21.18    |
| $\theta_{\rm sat}({\rm cm}^3{\rm cm}^{-3})$ | 0.534    | 0.552    | 0.561    | 0.501    | 0.574    |
| $K_{\rm sat}({\rm cmmin}^{-1})$             | 1.386    | 0.129    | 0.171    | 0.0363   | 5.737    |

coefficient of variation (CV%) of oven-dry bulk density,  $\rho_{\rm b}$ , is equal to 3.9%. Further, the average of the coefficients of variation computed for soil texture components (% sand, % silt, and % clay contents) is approximately 7.5%. Conversely, the CV% of organic C content is greater than the other soil properties and equal to 17.5%. Therefore, small variability in microporosity generated by primary particles (sand, silt, and clay), which characterize the "textural" behavior of the soil, should be expected.

5 On the other hand, large variability in the macroporosity should be found due to the larger statistical dispersion of the binding agent (organic C content), which favors the formation of aggregates. This feature can be the major reason for the observed significant variability in K<sub>sat</sub> (CV% equal to 163.3%). It is worth noting that such CV% value is typical for K<sub>sat</sub>, even at the small space scale of the hillslope plot (see (Ciollaro and Romano, 1995) among many others). The large variability in K<sub>sat</sub> values is also attributable to the relatively small ρ<sub>b</sub> measurements, which make the uppermost soil more prone to local
10 compaction and hence to disturbance of the aggregates.

The four undisturbed soil cores (0.07 m in length and 0.072 m in diameter) excavated at locations P02 to P05 are collected from a soil depth of 0.05 - 0.12 m. After estimating  $\theta_{sat}$  and  $K_{sat}$  as outlined above, the cores are reduced in height to approximately 0.04 m. Then, they are placed on a suction table for directly measuring soil water retention points (Romano et al., 2002). Soil water contents at greater suction head (h) values, ranging from 75 kPa to 1200 kPa, are determined using a series of pressure plate extractors (Dane and Hopmans, 2002).

- With regards to the undisturbed soil core collected at location P01, soil hydraulic properties are determined with the laboratory evaporation experiment. Measured variables are analyzed through a modified Wind's method (Peters and Durner, 2008) and using the optimization approach proposed by (Romano and Santini, 1999).
- Figure 3 shows the soil water retention,  $\theta(h)$ , and hydraulic conductivity  $K(\theta)$  functions (WRFs and HCFs, respectively) as 20 determined by the two laboratory methods presented above. WRFs for the undisturbed cores tested in the suction table present a similar behavior when approaching drier soil conditions. This is due to the relatively small variations in the soil textural properties (see Table 1). As discussed for Table 1, larger differences among the WRFs occur close to full saturation mainly due to the local and unpredictably different degrees of compaction and aggregation of the primary soil particles. Interestingly, the WRF pertaining to the soil core tested through the evaporation experiment shows a certain bimodal behavior, whose description
- through a bimodal relation leads to a much better prediction of K (as discussed in (Romano et al., 2012)), see the right panel of Figure 3.

5

In Figure 3, points obtained by processing measured data with a modified Wind's method match very well analytical findings obtained through the evaporation inverse method developed by (Romano and Santini, 2002). Importantly, HCFs of the soil cores collected at positions from P02 to P05 are simply predicted from WRFs parameters and from measured  $K_{sat}$  values. Differently, the HCF of the soil core collected at P01 is obtained simultaneously with the WRF from the measured variable during the evaporation experiments. The differences between the soil hydraulic properties of the cores collected at P02-P05 and those of the core collected at P01 can also be partially due to the different sizes of the cores (the core subjected to the evaporation experiment has a volume of about  $500 \text{ cm}^3$ , whereas the cores draining on the suction table have a volume of about  $250 \text{ cm}^3$ ). Further, differences in HCFs of cores collected at P02-P05 are emphasized by variabilities in the measured values of  $K_{sat}$ , which is very sensitive to local compaction conditions.

#### 10 2.2.2 Output discharge

Water exiting the experimental plot is collected in a v-shaped aluminum channel extending along the downstream side of the plot, see Figure 4(a). The channel discharges water and solids in an aluminum tank, where a water level sensor surveys output discharge rate. Such a tank is composed of three connected partitions, see Figure 4(b). Water enters the first section of the tank, where coarser solid material is collected, up to the height of the separating screen. In the second section of the tank, water

- 15 and finer sediments overflow from the first one. The last section of the tank is filled with water and solids entering from a rectangular aperture located at the bottom of the separating screen with the middle section. Water flows out of the tank through a v-notch weir section and is drained into a low-lying agricultural ditch. A Structural Testing System (S.T.S.) strain gauge detects the water level in the last partition of the aluminum tank.
- In Figure 5, the rating curve for the output discharge from the v-notch weir is presented. This relation is obtained by 20 performing experiments where known discharges are fed into the aluminum tank and water levels are measured with the S.T.S. strain gauge. Experimental data in Figure 5 are fitted to obtain the rating curve, where Q and h are the output discharge and water height above the v-notch weir, respectively. Specifically, red dots are experimental measurements; the blue dashed line is a fitted power law; the green dotted line is a fitted polynomial law; the magenta solid line is the rating curve for sharpcrested weirs with triangular control section ( $\alpha$  is the angle between the sides of the notch) (Bos, 1989) and effective discharge
- 25 coefficient ( $C_d$ ) function of the water head; and the dash-dotted black line is the rating curve for sharp-crested weirs with triangular control section with effective discharge coefficient equal to 0.6. The polynomial law accurately fits experimental data ( $R^2 = 0.99$ ).

#### 2.2.3 Turbidity

In addition to the water level sensor, the collection tank hosts a turbidity sensor probe. The probe allows for real-time measurements of the suspended solid material gathered in the final partition of the collection tank. Preliminary calibration experiments are performed to establish the relation between the suspended solid content and the nephelometric turbidity units (NTU) measured by the probe. Specifically, soil specimens are gradually added to a tank filled with 1001 of water. Each soil sample is

deployed at intervals of approximately 10 minutes and a total volume of 4.41 kg is loaded in the tank. In Figure 6, we report the linear relation between NTU units and the suspended solid content in the tank.

#### 2.2.4 Soil moisture

Soil moisture in the plot is monitored through four Campbell Scientific CS615 water content reflectometers. Such instruments can be oriented both vertically and horizontally to sense the water content in the soil. During experimental tests, the four probes are inserted in the working area of the hillslope plot as illustrated in Figure 7 to monitor soil moisture. Specifically, probes are installed after the soil was packed by digging soil pits. Measurement rods are thrust into the soil at 10 and 50 cm depths horizontally underneath the undisturbed region of the plot, thus causing minimal disturbance.

To estimate the water content sensed by the probes at saturation, in situ experimental tests are conducted. Specifically, water 10 is fed from the hillslope surface at the location of the probes for up to 24 hours to facilitate moisture migration. Saturation is achieved after a 24h-test for probes 2, 3, and 4. The upstream-deep probe gains a maximum value of 0.422; the upstreamshallow probe senses a maximum of 0.462; and the downstream-shallow probe attains a maximum of 0.364. The downstreamdeep probe achieves a maximum of 0.503 after a 21h-test.

To gain a preliminary insight on the illustrated soil hydraulic characterization and of the functioning of the soil moisture sensors, a simulation analysis of soil water in the unsaturated zone of the plot is performed using the HYDRUS-1D software package (Šimůnek et al., 2013). A simulation run is executed with reference to a rainfall event recorded from the 7th to the 13th of November 2014. In the simulation, the soil profile is assumed to be uniform and the soil hydraulic properties obtained through the evaporation experiment (see Figure 3) are utilized. The upper boundary conditions are the actual recorded precipitation and the evapotranspiration rates for a bare soil. At the lower boundary, a unit gradient of the total water potential head is imposed. The initial soil moisture conditions are those recorded by the DS and DD moisture probes.

Figure 8 presents the variations with time of soil moisture,  $\theta$ , as affected by the considered rainfall event. Comparisons between the measured and the corresponding simulated soil moisture values are also shown. The simulated soil moisture patterns at both depths follow very closely the measured data. Interestingly, the rise in soil moisture due to rainfall infiltration is fairly well described by the numerical model. On the other hand, major discrepancies between measured and simulated results occur in the subsequent phase, after the end of the rainfall event. This can be mainly attributed to the onset of some lateral flows in the deepest soil zone of the hillslope, which is not taken into account by the one-dimensional model.

Although soil water dynamics in the plot are complex, the 1-D simulation leads to realistic findings. Since soil hydraulic properties are determined independently (rather than calibrated) from the monitored data sets, the results depicted in Figure 8 confirm the goodness of the soil characterization performed in this study.

#### 30 2.2.5 Wetting front

25

Wetting front propagation is qualitatively monitored from glass panels located on a side of the plot. The 2m-large and 24mmthick panels are placed on the left side of the hillslope, see Figure 9, to allow for wetting front observations along the undisturbed working area. Estimations are conducted by periodically capturing pictures of the infiltration profile, which is easily

distinguishable due to the darker color of wet soil. Pictures are acquired with a GoPro Hero 2 set to high resolution and low acquisition frequencies. Direct reflections from sunlight and the detrimental effect of raindrops are accounted for by installing a platform roof above the glass panels. Figure 9 displays a picture of the infiltration profile acquired during a validation experiment. Wetting front propagation is inferred from observations of the side glass panels. Specifically, a GoPro Hero 2 camera is set at approximately 1 m from the central glass panel to capture high resolution pictures every 15 minutes.

## 2.2.6 Rainfall measurements

Rainfall is monitored through the available rain gauge system located in the same terrain parcel at 20 m from the hillslope plot. The rain gauging apparatus consists of a system of four tipping bucket collectors connected to a datalogger. Consistent with sensory apparatus on the hillslope plot, rainfall measurements are recorded every 5 minutes from each collector. The collectors

are evenly spaced at the corners of a  $10 \times 10 \text{ m}^2$  square and rainfall values averaged over the four recordings are used to estimate input precipitation.

#### 2.2.7 Fluorescent particle sensing system

The onset of surface runoff on the plot is studied using buoyant fluorescent particles as surface tracers. In particular, 0.98 g/cm<sup>3</sup> polyethylene beads purchased from Cospheric LLC are used to trace surface runoff. When exposed to UV light (365 nm 15 wavelength), the particles emit green light radiation (561 nm wavelength). We deploy selected amounts of fluorescent particles

- in the area illustrated in Figure 11, that is, in between the working region and the downstream 1.5 m-wide border. In this region, moisture is considerable as indicated by the rather dark soil coloring, and saturation is more likely to occur in short time periods. The beads are deployed on the hillslope at the start of rainfall, that is, at about 11:18 on January 30th, over a rectangular area extending for  $50 \times 30 \text{ cm}^2$ . The tracer particles are transported on the plot surface and then on the v-shaped
- aluminum channel located along the downstream side of the hillslope. They are further gathered on the surface of the central section of the collection tank where the detection unit is placed, see Figure 10. The unit consists of a case with an array of fourteen UV 8 W lamps in parallel and series connections. The case is waterproof and lamps are protected from water through a transparent plastic screen. A GoPro Hero 3 is used to monitor particle arrival at the collection tank. The camera is held in a waterproof case and is angled to capture the entire surface of the central section of the tank.
- As demonstrated in (Tauro et al., 2012b), groups of particles as small as 75 µm can be detected in case of rill flow on semi-natural hillslope plots. In the undisturbed working area, the soil cover does not present a rill-like structure or a preferential drainage network. Therefore, we investigate the initiation of runoff by selecting a mixture of particles of diverse diameters. Specifically, few grams of beads of diameters varying in the range from 75 to 1180 µm, are mixed with larger quantities of 710 850 µm spheres. The particle detection camera is set to 12 MP image resolution and to 0.1 Hz frame acquisition rate.
  Opaque panels are kept above the lamp case to minimize the effect of rain and direct light.

Particle arrival in time is surveyed by analyzing the image pixel coverage pertaining to the fluorescent beads. Such coverage is expected to increase as more particles are transported in the collection tank during the experiment. More precisely, we convert captured RGB frames to gray-scale and perform a morphological segmentation analysis Forsyth and Ponce (2002) to isolate

the beads from the rest of the objects depicted in the images. To this end, we process the frames with a filtering procedure that detects image edges. We further erode images with structuring elements of the shape and size of the fluorescent particles and reconstruct frames by connecting the only background pixels. We subtract such reconstructed images from the original frames to isolate the particles in the field of view. This procedure is performed on the entire sequence of approximately 700 frames recorded during the experiment. For each frame, we quantify the amount of pixels pertaining to the particles and normalize it with respect to the total pixel area of the frames to develop a breakthrough curve for the tracer.

5

#### **3** Results

### 3.1 Hillslope flow monitoring

- Observation of input rainfall and output water from the experimental plot leads to findings in Figure 12. In Figure 12(a), cyan
  bars show input rainfall measurements in mm/5min and the solid red line displays the output discharge estimated using the rating curve in Section 2.2.2 for the water level sensor. Data are reported up to a couple hours after the cessation of natural rainfall (14:00 on the last event on January, 30th). The experiment is performed during fairly heavy rainfall events. Specifically, precipitation is first observed in the afternoon on January, 29th. This event correspond to null output discharge from the plot. Other significant events are registered at night, leading to a high output discharge peak, and on January, 30th from 11:00 to
- 15 14:00. Such observations correspond to peaks in the output discharge delayed by a few minutes with respect to rainfall peaks. In Figure 12(b), the cumulated rainfall curve is depicted as a solid blue line and the cumulated output discharge as a solid red line. We note that less than half the total rainfall volume is discharged from the experimental plot in the analyzed time interval. The cumulated discharge curve is slightly shifted in time with respect to the cumulated rainfall.
- To further investigate the hydrological response of the hillslope to precipitation events, in Figure 13, we report soil moisture values recorded by the soil moisture probes. Specifically, measurements (W) are normalized by the values obtained at saturation ( $W_{sat}$ ) and presented in Section 2.2.4. Notably, the moisture conditions of the experimental plot are stable until 02:00, when a rather steep increase is observed at the two shallowest soil moisture probes, that is, DS in the downslope region and US in the uppermost region (refer to Figure 7 for probes naming). For the entire duration of the experiment, higher moisture is recorded from the deeper probe in the downslope region of the hillslope, and, before 02:00, driest conditions are found in the uppermost
- 25 region close to the soil surface. After 02:00, the shallower probes are the most affected by increasing precipitation and the downslope sensor records higher humidity with respect to the deeper one at approximately 13:00. The soil moisture behavior displayed in Figure 13 suggests that percolation to the deepest soil layers is expected to occur in rather extended periods of time, whereas quicker responses are observed on the plot surface. Specifically, a quantity close to half the total rainfall volume is rapidly discharged from the experimental plot. However, a greater part of precipitation both percolates through the soil and
- 30 is lost in evapotranspiration phenomena. Inspection of the soil through the glass panels did not result in a clearly detectable infiltration profile.

In Figure 14, we report turbidity measurements from the sensing probe and the estimated suspended solid volume in the last partition of the collection tank. The suspended solid volume is estimated by multiplying the turbidity in mg/l by the

output liquid volume measured in the last partition of the collection tank. Interestingly, a sharp increase in turbidity is observed around 12:00, that is, close to the peak of the last rainfall event. Turbidity consistently increases with rainfall until 13:30. After such time, the level in the collection tank decreases and part of the solid material is ejected, thus resulting in decreasing turbidity. As expected, the suspended solid volume closely follows precipitation regime in Figure 12(a) with peaks registered

- almost simultaneously with rainfall. These observations suggest that turbidity may be regarded as a proxy for runoff. Therefore, recorded output discharge is due to the contributing downslope area of the hill, where soil saturation rapidly occurs (Beven and Kirkby, 1979). In Figure 15, we report pictures of the plot before and after the experiment. Specifically, in the top image, a view of the hill is presented after the experiment, where the dashed box encloses the wetter soil of the downslope contributing area. In the bottom frames, the area where particles were deployed is illustrated at the onset and at the end of the experiment.
- In the right-side image, the darker soil color and the presence of small-scale rills indicate saturation and contribution to output runoff.

#### 3.2 Particle arrival distribution

In Figure 16, we report two snapshots recorded from the particle detection unit at the beginning and close to the end of the last observed rainfall event on January, 30th. Images depict the unprocessed original frames and the same images after
15 the segmentation procedure, where the only particles are depicted against an artificial homogeneous background. Fluorescent spheres tend to align at the borders of the tank and along water surface ripples. While the methodology shows good accuracy in detecting the particles against light reflections and different floating objects, edges between the water and the tank are not completely subtracted from the background image. However, the percentage of image coverage pertaining to the tank edges is constant during the experiment and, thus, does not affect particle identification. Notably, despite the small size of the particles,
they are clearly detectable and quantity considerably increases during the experiment.

In Figure 17, we report the time series of the particle image coverage for the experiment duration. Specifically,  $B_0$  indicates the number of pixels covered by the total amount of beads, whereas B denotes the number of pixels covered by particles in each image. We note that pictures are recorded up to about 1:15, that is, until the end of the last major rain event. Each marker in Figure 17 corresponds to a picture. Interestingly, the curve can be regarded as a standard tracer breakthrough curve,

- where the canonical S-shape is clearly distinguishable. In particular, particle arrivals at the detection unit are noted from the very beginning of the experiment. This is attributed to a number of factors; namely, particles are manually deployed from a height of 1 m from the soil surface, therefore, meager quantities of tracers are placed closer to the v-shaped aluminum channel due to wind effects. Further, rain-splash contributes to the particles' discontinuous displacement in the downslope direction. Starting from 12:24:30, the presence of particles in the tank increases sharply. From 12:57:50, the slope of the curve decreases
- sug