# Peer review of "Technical Note: Monitoring streamflow generation processes at Cape Fear"

_Hydrology and Earth System Sciences, 2016_

## Referee Comment (RC1) · Anonymous Referee #1 · 23 Oct 2016

The authors describe a sophisticated monitoring system in an artificial slope of dimensions 7 m (long) by 7 m (wide). Based on the stated volume (40 m3), the average soil depth was 0.81 m, although it is not stated whether this depth was uniform. The impermeable barrier at the base of the plot and the borders along the sides of the plot, impose artificial boundary conditions which facilitate simplifying the hydrologic conditions and controlling water fluxes for basic understanding of hillslope hydrological processes, but (on the other hand) are a bit unrealistic except for artificial slopes and very homogeneous soils. The objective of the paper, although not specifically stated, appears to be to study the hydrological response to external rainfall through this "proof-of-concept" experiment. Results from only one storm sequence are presented, thus I assume that the thrust of the paper is on the uniqueness of this proof-of-concept methodology.

While the experimental setup is a bit unique, I take issue that this could be considered

a "proof-of-concept". Interestingly, in the Introduction, the authors mostly describe recent hillslope plot studies (Biosphere 2 Hillslope; Hydrohill; etc.), but ignore important earlier work. Specifically, they do not mention the classic Hewlett & Hibbert plot study in the Coweeta Experimental Watershed – results were published in the early 1960's and formed the basis for the 'variable source area' concept for streamflow generation. Furthermore, there have been a many studies (dating back into at least the 1970's) conducted in large flumes and constructed hillslopes to assess the interactions of pore water pressure generation and slope stability. Given all of this previous work in constructed hillslopes, I fail to see how the study described in this paper fits the definition of a 'proof-of-concept'.

Based on my assessment, I suggest that the authors conduct a series of unique hydrologic experiments in this artificial hillslope. Being able to capture natural rain events is a definite advantage. The fluorescent dye tracing can be used as described to examine the initiation of erosion features for different cover conditions, but this will take multiple experiments. In short, there is good potential here to utilise this experimental setup to examine some basic surface hydrology and erosion phenomena, but describing just one storm sequence is insufficient for a publication in HESS.

Other Comments:

Introduction: I have no idea why the authors selected the references that are listed as examples of hydrologic processes in natural and artificial slopes. Many seem inappropriate to the discussion, and only a few are early important works. The statement on page 2, lines 20-21 appears to be a quote, but makes little contribution to the discussion. The statement on page 3, lines 7-8 is incorrect – just because the system is exposed to natural events does not mean it allows for establishment of hydrologic processes found in natural ecosystems. The disturbed nature of soil (i.e., no or limited natural soil structure) in the plot is very different from actual hillslope soils.

Section 2.1: I would not categorize the January 29-30 storm as two events given their

proximity – this is really only one storm sequence.

Section 2.2: When and how was the soil placed into this structure? Was the soil left to 'settle' for a long time so that some minimal soil structure could begin to form? Was the soil compacted in layers as it was put into the plot? Was the depth the same throughout the plot (0.81 m)? Make it clear that the lower end of the slope was open and allowed to freely drain (or is this not the case)? Assuming this is the case, you should note that this imposes an artificial boundary at the lower slope (i.e., water will not drain until soil is saturated). This could promote saturated overland flow.

Section 2.2.1 Why not try to estimate Ksat from in-situ measurements rather than small cores? You did not show the number of samples analysed to calculate variability (page 4, line 33). Ksat values are typically log-normally distributed, thus conventional CV's are not really appropriate. The statement on Page 5, Lines 9-10 is confusing: you note that the low bulk density of the soil makes it prone to compaction and disturbance of aggregates. I agree about the soil being prone to compaction, but I doubt that this disturbed soil had well-formed aggregates (see my previous comment). On page 5, line 16, please state the specific hydraulic properties you are referring to.

Section 2.2.2 The purpose of the third tank is a bit unclear. A reference is needed for the S.T.S. strain gauge.

Section 2.2.4 Why did you use a 1-D model to simulate flow; of course there should be lateral flow unless the surface conditions were such that most of the rainfall moved downslope as overland flow, and this appears not to be the case.

Section 2.2.7 The statement on Page 8, lines 26-27 is unclear – please rephrase.

Section 3.1 Pg. 9, lines 21-25: These results are completely predictable and widely known. Pg. 9, line 33: turbity is in NTU; suspended sediment is in mg/L Pg. 10, lines 4-11: It is not clear how the plot configuration may have affected lower slope drainage and the build-up of a water table; see earlier comments. Section 3.2 Page 10, Lines 13-

33: These two paragraphs describe issues dealing with "Methods" rather than process understanding derived from results. Page 11, lines 1-6: Do you mean that no rills were present on the soil surface at the onset of the experiment (line 1)? What do you mean by saying "particle transport becomes more regular"? By "remarkable recovery", are you referring to the 97% recovery?

Section 4 Page 11, lines 8-15: This reiterates the homogeneous conditions, but nothing new is described. Quick response to rainfall is not a new finding. Page 11, line 19: I assume you mean preferential surface pathways – please note that these can lead to rill formation. Page 11, lines 20-21: But Dunne mostly worked in field settings. You never really describe the antecedent soil moisture conditions. Page 12, lines 1-2: I would argue that the specific hydrologic processes that you can derive from plot studies with uniform soils are mostly known. Page 12, lines 8-9: If you want to examine the effects of soil structure and vegetation over time in a single experimental plot like Cape Fear, this would take many years. Page 12, lines 12-14: If you sub-divide the plot, then you would have quite small plots to work with and would lose the concept of a hillslope experiment.

Section 5 The Conclusions are quite obvious.

---

## Short Comment (SC1) · 16 Nov 2016

We thank Referee #1 for the detailed review of our work, and would take this opportunity to make clearer our project scope and to provide some preliminary responses to the major comments raised by the Referee.

The main objectives of the present technical note are to present an experimental "hybrid" hillslope plot and to assess whether or not this plot is adequate for studying basic hydrological processes (Page 3, Lines 9-10: "In this paper, we assess the feasibility of studying streamflow generation processes in the "intermediate" experimental settings of Cape Fear"). The suitability to investigate hydrologic phenomena is assessed by evaluating the response of Cape Fear to precipitation events. The described proof-of-concept experiment does not aim at discovering new knowledge. However, despite

inevitable limitations imposed by limited natural soil structure, artificial boundary conditions, homogeneous soil etc., the well-known response of the setup suggests that the plot can be adopted to examine basic hydrologic phenomena. The proof-of-concept experiment presented in this tech-note is not employed to deeply unravel streamflow generation processes. Conversely, in the future, comprehensive studies, including several replicates of each experiment, will be carried out by using fluorescent dye tracing and camera systems to address specific research questions. To better highlight the scope of our work, in the new version of the manuscript, we will more clearly state the objective and modify the title to "Technical Note: Monitoring hydrological processes at Cape Fear".

We deem the presented material pertinent to the requirements of HESS Technical Notes to "report new developments [. . .] and novel aspects of experimental [. . .] methods and techniques which are relevant for scientific investigations within the journal scope". As acknowledged by Referee #1, in the manuscript, it is shown that there is "good potential [. . .] to utilize this experimental setup to examine some basic surface hydrology and erosion phenomena". Therefore, the manuscript should be in line with the scope of HESS Technical Notes.

We agree with Referee #1 that the literature cited in the Introduction can be considerably improved. In the current version of the paper, we have focused on "hybrid hillslope" studies, that are closely related to our setup. We plan on including important contributions, such as publications on the "variable source area" concept as well as on interactions of pore water pressure generation and slope stability, in the new version of the manuscript.

With specific regards to the proof-of-concept experiment, Figure 13 depicts the normalized water content measured by the soil moisture probes. The probes located in the downstream area of the plot (DS and DD) monitored higher values of soil-water content than the upstream probes. This was consistently found throughout the year. In particular, in the five days preceding the experiment, recorded water contents are

within 3% of values relative to January, 29th at 10 am. We will include a comment on antecedent soil moisture conditions in the revised version of the paper. In response to rainfall, water content sharply increases in the downstream shallow soil layer and it slowly raises in the deeper layer. As expected, this behavior suggests the occurrence of saturation excess overland flow and, therefore, the feasibility of observing basic drainage formation in the plot.

While we will address each comment raised by Referee #1 in detail upon receiving all reviews, we would like to emphasize that future studies will focus on the effect of diverse surface covers on basic surface hydrology. We thank Referee #1 for advising that dealing with rather small sub-plots may not be representative of the concept of hillslope experiment. We will opportunely design the experimental setup and present multiple replicates for each test in future contributions.

---

## Referee Comment (RC2) · Anonymous Referee #2 · 20 Dec 2016

The manuscript of Tauro et al. presents an artificial hillslope plot equipped with different devices for the measurement of output discharge, soil moisture, surface runoff, rainfall and sediment load. The aim of this paper is to show that the monitoring system is suited to study runoff generation and related processes at the hillslope scale. To this end, phenomena and fluxes observed during a natural storm event were presented and discussed. As a whole, the manuscript addresses relevant scientific questions within the scope of HESS. However, in my opinion, the Ms is in its current version not suitable for publication. Artificial hillslope plots are well-known. Thus, I believe that discussing the feasibility of the experimental setup for studying runoff generation processes is not enough for getting published in HESS. In my opinion, the paper would have a larger scientific impact if the authors show how they use their experimental design to address scientific questions as outlined at the end of the Ms (page 12, 5-10).

Other comments 1. State of the art of artificial hillslope plots is not well covered in the introduction. Artificial hillslope plots are well-known. For instance, I have seen many of such plots in different regions of China. So, it needs to be shown what is new about the presented approach. 2. The structure of chapter 2 should be revised. (a) plot description, (b) devices/ measurements, (c) lab measurements, (d) specific field experiments (e.g. storm event). The rainfall simulator should be described as well. 3. Why do you not use HYDRUS 2/3D for modelling (page 7, 15-25)? Much more could be learned by combining your measurements with 2/3D modelling! The geometry of your plot, the upper and lower boundary condition as well as the outflow is well known. 4. I know the budget is always limited but if feasible I would recommend installing full-range tensiometer at different depths and positions. It would significantly improve your knowledge about the role of your soil in runoff generation. 5. You may do an irrigation experiment for the inverse determination of the soil hydraulic properties of your artificial plot (the plot needs to be covered by a plastic cover after irrigation). As mentioned before, soil pressure head measurements in combination with 2/3D Hydrus modelling would make that feasible. An evaporation experiment according to Wind might be also possible if you could install more soil moisture probes for accurate balancing of the evaporative fluxes.

---

## Author Comment (AC1) · 17 Jan 2017

We thank Referee #2 for the review of our work. Here, we clarify several issues and provide responses to major comments raised by the Referee.

We are grateful that Referee #2 correctly stated the objective of our manuscript. In our opinion, our work is in line with the requirements of HESS Technical Notes and is expected to help experimental hydrologists by describing novel methods and techniques for scientific observations. In the following we reply to the specific points raised by Referee #2.

1. Artificial catchments are far from being a well-known topic and several research groups worldwide are building their own experimental setups to test hydrological hypotheses and questions. Most frequently, relatively small plots or flumes are routinely

established at the laboratory scale to respond to simplified questions. More comprehensive "hybrid" outdoor setups, such as the one presented in our work, that interact with external natural agents, are more rarely designed and reported in the literature. We agree with Referee #2 that the literature cited in the Introduction could be improved upon his/her suggestion of specific studies related to our work. However, we remark that we have done our best to cover the existing literature on the subject, and Referee #2 did not provide any additional reference to previous work on artificial hillslopes. Thus, in variance with the Referee's statement, we do believe that the topic is not settled yet, being still a matter of research. Our manuscript for HESS Tech-note has also the scope of forming a contribution to a discussion on this topic while providing some preliminary outcomes.

2. We agree with Referee #2 that Section 2 on Materials and Methods could be somewhat restructured according to his/her suggestions. We will be happy to include a thorough characterization of the rainfall simulators which was left out from the original submission for the sake of brevity.

3. With regards to HYDRUS-1D simulation, the analysis was conducted to provide simple but sound insights on soil functioning at Cape Fear, not a thorough simulation of water flow in the soil. Such analysis provided a benchmark to test the reliability of the soil response recorded by the sensors and the accuracy of the laboratory-based soil hydraulic characterization. We will better point out the objectives of the analysis in the revised version of the manuscript.

4. We agree with Referee #2 that much more could be learned by combining our measurements with modelling with HYDRUS 2/3D. Indeed, the preliminary soil characterization was purposely conducted in the laboratory on soil samples. Higher dimension investigation of Cape Fear soil response is the objective of future studies.

5. As correctly suggested by Referee #2, future studies will also aim at conducting a hillslope-scale evaporation experiment. While these research directions are certainly

interesting, in our opinion, they deserve dedicated analyses and contributions, well beyond the present Technical Note.